# Variability in Codon Usage in Coronaviruses Is Mainly Driven by Mutational Bias and Selective Constraints on CpG Dinucleotide

**DOI:** 10.3390/v13091800

**Published:** 2021-09-10

**Authors:** Josquin Daron, Ignacio G. Bravo

**Affiliations:** 1Laboratoire MIVEGEC (CNRS, IRD, Université de Montpellier), 34394 Montpellier, France; Ignacio.bravo@cnrs.fr; 2Center for Research on the Ecology and Evolution of Diseases (CREES), 34394 Montpellier, France

**Keywords:** virus evolution, SARS-CoV2, host switch, codon usage bias

## Abstract

The *Severe acute respiratory syndrome coronavirus 2* (SARS-CoV-2) is the third human-emerged virus of the 21st century from the *Coronaviridae* family, causing the ongoing coronavirus disease 2019 (COVID-19) pandemic. Due to the high zoonotic potential of coronaviruses, it is critical to unravel their evolutionary history of host species breadth, host-switch potential, adaptation and emergence, to identify viruses posing a pandemic risk in humans. We present here a comprehensive analysis of the composition and codon usage bias of the 82 *Orthocoronavirinae* members, infecting 47 different avian and mammalian hosts. Our results clearly establish that synonymous codon usage varies widely among viruses, is only weakly dependent on their primary host, and is dominated by mutational bias towards AU-enrichment and by CpG avoidance. Indeed, variation in GC3 explains around 34%, while variation in CpG frequency explains around 14% of total variation in codon usage bias. Further insight on the mutational equilibrium within *Orthocoronavirinae* revealed that most coronavirus genomes are close to their neutral equilibrium, the exception being the three recently infecting human coronaviruses, which lie further away from the mutational equilibrium than their endemic human coronavirus counterparts. Finally, our results suggest that, while replicating in humans, SARS-CoV-2 is slowly becoming AU-richer, likely until attaining a new mutational equilibrium.

## 1. Introduction

The *Severe acute respiratory syndrome coronavirus 2* (SARS-CoV-2) is the cause of respiratory disease COVID-19, occasionally leading to acute respiratory distress syndrome and eventually death [1]. With no antiviral drugs nor vaccines initially available, and with the presence of asymptomatic carriers, the COVID-19 outbreak turned into a public health emergency of international concern. Before the 2019 zoonotic spillover, viruses closely related to SARS-CoV-2 had circulated probably for decades in bats as well as in other intermediate hosts, such as the Sunda pangolin *Manis javanica* [2]. Cross-species transmissions are common among coronaviruses (CoVs) [3,4], and they are an important mechanism driving the evolution of bat-CoVs in nature [5,6]. In humans, all CoVs are likely linked to zoonotic events, mostly from bats or rodents, with occasionally domestic animals playing the role of intermediate hosts [7]. Consequently, given the high zoonotic potential of the *Orthocoronavirinae* [8,9] and the vast repertoire of mammalian and avian hosts they infect, there is an urgent need to evaluate the potential zoonotic risk for each individual CoV. This knowledge will guide virus discovery, surveillance and research to identify for each virus the differential risk of efficiently infecting humans and cause a new pandemic.

None of the viruses infecting vertebrates encodes for any tRNA nor for any ribosomal protein, and CoVs are no exception. As a result, the translation of viral proteins relies exclusively on the host tRNA repertoire and translational machinery [10]. In order to efficiently support the production of viral proteins, it has been proposed that viruses would evolve to use the set of synonymous codons found overrepresented in their hosts, as the result of an adaptation to their host cellular environment [11,12,13,14]. This hypothesis is based on the fascinating discovery that codon usage is subject to natural selection [15,16,17,18,19]. This adaptive hypothesis, called translational selection, proposes that the non-random usage of synonymous codons and the abundance of tRNAs have co-evolved to optimize translation efficiency. Experimental analyses have been successful at characterising how syno-nymous substitutions influence the cellular fitness of an organism, by acting on a broad range of cellular processes, including changes in transcription [20], translation initiation [21,22], translation elongation [23], translation accuracy [17,24], RNA stability [25], and splicing [26].

Numerous studies conducted essentially in certain phages and their bacterial hosts have reported a strong match between the codon usage bias (CUB) of viral genes with respect with their hosts [27,28,29]. Similarly, the CUB of human papillomaviruses has been associated with differential clinical presentations of the infections [30], with a stronger virus–host match in CUB for human papillomaviruses causing productive lesions than for those causing asymptomatic infections. However, such a trend has not been observed for SARS-CoV-2: most of the highly frequent codons in SARS-CoV-2 are A- and U-ending codons [31,32] overall showing a poor match with the average human CUB [33]. This mismatch has raised important questions about the nature of the CUB in CoVs, the underlying mechanisms involved, and the impact on virus longer-term evolution. Furthermore, investigating the CUB match between viral and cellular genes constitutes a challenge to supporting the use of synonymous mutations as biotechnological tools to develop live attenuated vaccines [34].

In addition to being driven by constraints on tRNA abundance, the non-random usage of synonymous codons in viruses is also shaped by dinucleotide abundance. It has been long established that many RNA viruses infecting mammals have evolved CpG and UpA deficiency [35,36,37,38]. In the mammalian RNA Echovirus 7, it has been formally demonstrated that the severe attenuation of viral replication can be attributed to the increase in CpG and UpA dinucleotides frequencies rather than to the use of disfavored codon pairs [39]. Viruses with high CpG or UpA frequencies may be more likely to be recognized by pathogen innate immune sensors, preventing them from replication initiation [40]. Functionally, the inhibition of viral replication and the degradation of the viral genome have been attributed to the zinc finger antiviral protein (ZAP), a powerful restriction factor that specifically binds to CpG motifs [39]. It is hence critical when investigating CUB to disentangle confounding effects of CUB and dinucleotide abundance.

Finally, non-adaptive evolutionary forces can also affect the viral genome composition. Because fitness differences associated with individual codons are very small, it requires very large population sizes (i.e., in the case of viruses highly productive and highly prevalent viral infections) for natural selection to act and lead to a significant impact on the global genomic CUB. This trend is verified in large organisms with small population sizes, such as most mammals, where natural selection on CUB is weak [41,42] and CUB is instead primarily shaped by mutational biases [43] and GC-biased gene conversion [44]. Mutations are the fundamental substrate for genotypic diversity, leading to phenotypic diversity upon which natural selection can act. Point mutations are stochastic processes but they concur with certain deterministic and directional biases, in bacteria [45] as well as in eukaryotes [46,47,48,49]. Previous studies suggest that mutations occurring during genome replication are universally biased towards AT. Similarly, for SARS-CoV-2, analysis of mutational profiles indicates a strong mutation bias towards U [50,51]; however, the impact of this bias on CUB variation has not been characterized yet at the scale of the whole *Orthocoronavirinae*.

Given the impact of the match between virus and host CUB in viral gene expression, the key challenge is to determine the impact of a specific genome composition or CUB in the initial zoonotic spillover from animals towards humans that may eventually modulate the risk of stable human-to-human transmission. Here, we investigated the CUB variability in *Orthocoronavirinae*, with an emphasis on CoVs infecting humans. We aimed at determining whether CUB in CoVs is actively selected according to the codon preferences of their hosts or whether it reflects instead the action of other evolutionary forces. We show that variation in CUB in *Orthocoronavirinae* mainly results from differences in GC3-content and in CpG and UpA abundance, independently of the host infected. Variation in GC3-content is primarily dictated by mutation biased towards AU, a trend universally observed in all *Coronavirinae* genera, regardless of the host. Finally, selection against CpG and UpA dinucleotides strongly impacts the CUB of CoVs. Altogether, we conclude that variation in CUB plays a minor role, if any, on the probability of the establishment of a zoonotic spillover towards humans.

## 2. Results

### 2.1. Variation in Codon Usage Bias among Orthocoronavirinae Is Not Dependent on the Host

To better understand the causes of CUB differences between CoVs, we investigated a total of 82 complete CoVs genomes. Our sample spans viral and host taxonomical diversity, covering the four viral genera within *Orthocoronavirinae*, and embracing a total of 47 different vertebrate host species within nine mammalian and five avian orders (Appendix A). First, we explored the variation in the 59 synonymous codons frequencies through a Principal Component Analysis (PCA) (Figure 1A, Appendix A). The PCA efficiently reduced information dimensionality as the first two components captured, respectively, 34.9% and 20.1% of the variance, and the first four dimensions contributed with above 70% of explanatory power. Interestingly, the 82 CoVs were distributed scattered along the two first PCA axes without any obvious stratification as a function neither of the viral taxonomy nor the host infected. This result contradicts our initial hypothesis of a host-specific CUB signature in CoVs and suggests that translational selection for convergence towards the host’s CUB is presumably weak.

The PCA results contribute further with information about the spreading of the individual codons as well as their contribution to the total variance. The first PCA axis sharply splits the codons depending on their nucleotide in the third codon position (often referred to as GC3), with the exception of UUG-Leu codon, which stands alone among all other A- or U-ending codons (Appendix A). Strikingly, we found that variation in genomic GC3 content strongly correlated with variation on the first PCA axis (adjusted R^2^ = 0.94, *p* value < 1 × 10^−50^, Figure 1B). Note that variation in GC3 did not show any correlation with any other of the main PCA axes (Appendix A). In-depth analysis of the frequency patterns for the 18 synonymous codons families showed that A- or U-ending codons are systematically preferred over the G- or C-ending ones (Figure 1C). This trend was especially true for amino acids with multiplicity two (i.e., encoded by two codons), but also confirmed for amino acids with multiplicity four: U-ending codons were systematically the most used among the four codons, while A-ending codons were preferred over the G-ending ones. For amino acids with multiplicity six, the overall scheme corresponds to the combined patterns of a family of multiplicity four and a family of multiplicity two. Altogether, our observations show that variation in GC3 is the main explanatory factor for CUB variation between CoVs, which strongly suggests a universal AT-ending over GC-ending synonymous codons enrichment in CoVs genomes.

We aimed at further quantifying the proportion of the global variability in CUB that is explained by the host and by the virus taxonomic stratifications. Through a correspondence analysis, we associated a subset of 75 different viruses of our codon usage table into blocks corresponding to the seven host taxonomic orders (only host taxonomic orders with at least five viruses were considered; Appendix A), allowing us to split the total variability into between-category and within-category variability. The top 10 eigenvalues of this decomposition are represented in Figure 1D showing that within-host differences in CUB explain four times more of the overall variability than differences between-hosts (respectively 76.3% vs. 23.7%); the explanatory power of both levels is larger than the randomly expected one from a homogeneous distribution (8.1%, *p* value < 9 × 10^−4^). A similar analysis was reproduced associating the viruses into blocks corresponding to the four viral genera within *Coronavirinae*. We observed that within-genus differences in CUB explain over five times more of the overall variability than between-genera differences (respectively 85.6% vs. 14.4%); again, the explanatory power of both levels is larger than the randomly expected one from a homogeneous distribution (3.7%, *p* value < 9 × 10^−4^). Together, our results are consistent with our initial PCA analysis, and suggest that both host and virus taxonomy variables are not the main factors that drive variation in CUB within *Orthocoronavirinae*.

### 2.2. The Mutational Spectrum in Orthocoronavirinae Is AU-Biased

Since variation in GC3 content is the main individual driver of the variation in CUB between CoVs, we investigated next whether mutational biases are the underlying main force driving nucleotide content. Population genetics studies show that in order to explore mutational biases independently of the effect of natural selection, one should work at shallow, short-term evolutionary periods, where natural selection is less powerful to impact nucleotide polymorphism patterns [45]. Consequently, we estimated the frequencies of individual transition and transversions by narrowing down our analysis to a subset of 16 different CoV metapopulations. From the public databases (see Methods), we downloaded a total of 12,102 different viral genome entries to construct these 16 different metapopulations (Appendix A). Prior to the identification of SNPs for each metapopulation, we carefully preprocessed each dataset by removing putative recombinant sequences (see Methods, Appendix A), as recombinant sequences violate the assumptions for phylogenetic inference methods used downstream. For each viral metapopulation, we removed further the effect of population structure by phylogeny-driven selection of one single homogeneous population per viral species (Appendix A). Our final dataset was composed of a total of 10,373 viral genomes for the 16 metapopulations (median number of genomes per metapopulation 31; range 9–9588), for which a total of 57,059 SNPs were called (median number of SNPs called per metapopulation 3037; range 253–7778; Appendix A). Despite the heterogeneous sampling size across our metapopulations, the number of SNPs called was highly comparable, the exceptions being SARS-CoV-1 sequences from humans and MERS sequences from humans and from camels, which exhibited a low number of sequence polymorphisms, albeit large enough to grant sound analyses (respectively 716, 621 and 253). There was no obvious correlation between the number of genomes in each metapopulation and the number of sequence polymorphisms retrieved (R^2^ = 0.183, F_(1,14)_ = 3.14, *p* = 0.098), even less when removing the SARS-CoV-2 metapopulation (R^2^ = 0.054, F_(1,13)_ = 0.74, *p* = 0.405), which was an outlier in number of genomes considered (n = 9588).

Under a maximum likelihood framework, we separately estimated for the synonymous and non-synonymous compartments the frequencies of transitions and transversions using the generalized time-reversible model for the phylogenetic reconstruction fitting the genomic data best. This analysis allows us to estimate the frequencies of individual transitions and transversions after correcting for nucleotide composition in each compartment. Our results show that for the synonymous substitutions compartment and for all viral metapopulations studied, U<>C transitions were more frequent than A<>G transitions, and for most metapopulations C->U transitions were more frequent than the reverse U->C (Figure 2A). These remarkable differences were of a larger magnitude for substitutions occurring within the synonymous compartment than within non-synonymous compartment, where the C->U changes were only slightly more common than the reverse substitution. Consistent with these findings, stratification of substitutions into GC-enriching (i.e., AU->GC) or AU-enriching (i.e., GC->AU) categories shows that for all metapopulations AU-enriching substitutions occur at much higher rate than GC-enriching ones in both synonymous and nonsynonymous compartments (respectively mean fold change = 2.49, *p*-value < 7.6 × 10^−6^; and mean fold change = 1.38, *p*-value < 7.6 × 10^−6^; paired Wilcoxon signed rank test). Individual departures from this trend are observed and may deserve future, focused research, the most obvious case being the avian gamma-coronavirus *Infectious bronchitis virus*, for which G->A and U->C transitions predominate. Altogether our results suggest that the mutational spectrum in CoVs is biased towards AU-enrichment and that the nucleotide content of the viral genomes is primarily determined by mutational biases.

### 2.3. Recent Human Coronaviruses Display Greater Mutational Disequilibrium Than Endemic Human Coronaviruses

Humans are the host to endemic CoVs, responsible for common respiratory diseases, but it has been proposed that these endemic hCoVs have an ancient zoonotic origin. Molecular dating studies have tried to evaluate the timing for host shift events for endemic hCoVs, estimating an old emergence of several hundreds of years ago for NL63 and a more recent one in the 19th–20th century for 229E, OC42 and HKU1 [52,53,54,55]. This epidemiological stratification among hCoVs, differentiating recently zoonotic hCoVs and ancient zoonosis that have become endemic in humans, provided us with the unique opportunity to characterize the impact that spillover and subsequent establishment in the new host has on the mutational equilibrium of the virus genomes. Previous studies have demonstrated that a high-resolution estimation of the mutational equilibrium can be gained from the simple property of the folded site frequency spectrum (SFS) of AU-to-GC mutations, in which SNPs are not polarized [56]. The theory proposes that for populations at demographic equilibrium the GC-content converges towards the mutational equilibrium, yielding a perfect U-shape in the AU-GC allele frequency distribution, resulting in a symmetrically folded SFS. Deviations from the U-shape distribution would thus reflect that the genomic GC-content is not at its mutational equilibrium, a negative (or positive) skewness indicating higher (or lower) GC-content than expected under the mutational equilibrium. We applied this framework, computed the folded SFS for the seven hCoVs species (Figure 2A) and quantified the deviation from the expected U-shape distribution by calculating the skewness of each folded SFS as a proxy for the departure of the GC-content from the expected mutational equilibrium (Figure 2B). Our results show that for all four endemic hCoVs, the observed skewness was not different from the null expectation, while for the three recently emerged hCoVs (SARS-CoV-1, MERS-CoV and SARS-CoV-2) we observed a significant departure from the expected GC content at equilibrium. Both MERS-CoV and SARS-CoV-2 displayed a negative skewness, meaning that their genomes were slightly more GC-rich than the anticipated mutational equilibrium, while SARS-CoV-1 displayed a positive skewness, reflecting an AU-richer genomic content than expected for the mutational equilibrium. Because the SFS represents population summary statistics, potentially sensitive to population size changes (growth or contraction), we aimed to determine whether the exponential growth of SARS-CoV2 was modulating our estimation of the SFS-skewness. Interestingly, we did not observe any impact of the exponential growth of SARS-CoV2 on the estimation of the skewness (Appendix A). While the exponential growth of SARS-CoV2 changes the shape of the SFS by increasing the proportion of rare variants, this should symmetrically affect AU to GC and GC to AU mutations. Consequently, as theoretically demonstrated by Glemin and coworkers [56], the result of an SFS skewness value significantly different from the null expectation is robust to demographical changes.

Finally, benefiting from the wealth of sequence data generated on SARS-CoV-2, we investigated the dynamics of the genomic GC-content over the spread of the COVID-19 pandemic (Figure 2C). We observed that the GC-content of SARS-CoV-2 is very slowly albeit significantly decreasing with the progression of the pandemic (Figure 2C, R = −0.11, *p*-value < 1 × 10^−49^). This result of a trend towards an overall AU-enrichment is consistent with the SFS observation of a viral genome far from compositional equilibrium and suggests that during the expansion of SARS-CoV-2 in humans, the viral population is experiencing a GC->AU mutational bias resulting in a slow decrease in the GC genomic content compared to the viral genomes retrieved from humans at the origin of the pandemic.

### 2.4. CpG Dinucleotides Are Selected against in Orthocoronavirinae Genomes

Our initial characterization demonstrated that variation in GC3 content is the prevailing force shaping CUB among CoVs, explaining one-third of the total variation in CUB. We thus tried to identify other evolutionary forces further shaping CUB in CoVs. Previous works have identified that in several RNA viruses infecting humans certain dinucleotides are under-represented, notably CpG and UpA, and that this low dinucleotide frequency has a strong impact on codon-pair bias [57]. We investigated thus the ratio of the observed over the expected dinucleotide frequency in the CoVs coding sequences. Figure 3A shows that the observed abundance of the CpG dinucleotide in all CoVs lineages is significantly lower than the null expectation based on the individual nucleotide frequency, while this is not the case for the UpA dinucleotide. To a lesser extent, we identify CpA and UpG to be marginally more frequent than expected, which can be linked to the strong decrease in CpG, as they correspond, respectively, to the transitions CpG->CpA and CpG->UpG. We further state that the observed/expected ratios for CpG and UpA correlated well with the second and third dimension of our PCA analysis, respectively (R^2^ adjusted = 0.67 and 0.44; Figure 3B,C). It is important to remember that these dinucleotide frequency values have been estimated for the complete viral coding sequence and are not limited to the codons themselves, i.e., we also considered the presence of CpG and UpA dinucleotides in the codon boundary context, so that this impact is not simply related to higher frequency of CG-rich or AT-rich codons. This specific point will be addressed below. Overall, our results show that variation in CUB between CoVs is associated with variation in CpG and UpA dinucleotide content.

In order to formally demonstrate the impact that CpG and UpA dinucleotide frequencies have on CUB, we compared the synonymous codon frequencies among codon families either containing or lacking a CpG- or UpA-ending codon. First, only the Arginine amino acid allows for synonymous mutations between the first and second codon position that lead to CpG changes, as it is encoded by CGN and by AGY. A CGY->AGY transition thus allows for a synonymous substitution that results in the loss of a CpG motif. Such a depletion is present in the human genome, in which ca. 42.6% of the Arginine residues are encoded by AGY, but is cogent in the SARS-CoV-2 genome, where this value amounts to ca. 57.8%, the third highest value among the analysed CoVs genomes (Appendix A). This value is particularly high when compared to other human-infecting CoVs, as it reaches 49.7% for the zoonotic SARS-CoV-1 genome, and amounts only to between 47.7% and 37.7% for the four endemic human CoVs (Appendix A).

Second, we focused on the synonymous changes between second and third codon position. Having previously established that mutational biases modulate frequencies among synonymous codons, we accounted for this confounding effect by calculating an expected ratio in synonymous codon frequencies for the codon pairs in the form of XXA/XXG. To this end, we defined as our reference set for comparison the three codon families with multiplicity two lacking CpG and UpA dinucleotides and ending by either A or G (i.e., Gln-CAA/G-, Lys-AAA/G- and Glu-GAA/G-, indicated as “XXpA/XXpG” in Figure 3D). For this reference set, we calculated an overall fold change XXA/XXG ratio of 1.26, consistent with the AT-enriching mutational bias described above. For amino acids encoded by CpG-ending codons (Ser4, Pro, Thr, Ala, indicated as “XCpA/XCpG” in Figure 3D), we observed a 5.33-fold change XCA/XCG ratio, significantly higher than the corresponding one for the reference amino acids set (within-genome paired Wilcoxon signed rank test, *p*-value < 1 × 10^−14^). This difference indicates that regardless of the amino acid encoded, CpA-ending codons are systematically preferred over their CpG-ending synonymous counterparts, at higher proportion than expected under the pressure of A->G mutational bias alone. In the case of UpA-ending codons (Leu2, Leu4, Val), we observed an overall XXA/XXG ratio of 1.04, slightly but statistically significantly lower than the corresponding one for the reference amino acid set (within-genome paired Wilcoxon signed rank test, *p*-value < 1 × 10^−6^). Thus, we interpret that although the UpA-dinucleotide is not significantly depleted in CoV genomes, UpA-ending codons are less frequent than their synonymous UpG counterparts. Consequently, the genomic UpA dinucleotide-content is shaped by two antagonistic evolutionary forces: on the one hand a mutational bias promoting the overall excess of G->A transitions and on the other hand a selection for XUpA->XUpG transitions in synonymous codons. Indeed, this interpretation also supports the finding that the UpG dinucleotide is borderline significantly more frequent than expected, as it also corresponds to the transition UpA->UpG.

Finally, we aimed at characterizing the impact that the pressure against CpG and UpA dinucleotides imposes on CUB, when these dinucleotides are located at the codon pair boundary, such as XXC-GXX or XXU-AXX. We thus computed the frequency ratio XXU/XXC for synonymous codons depending on the nature on the downstream first nucleotide: if a selection again CpG nucleotides existed, we would expect the XXU/XXC ratio to be higher when the downstream codon starts with G; similarly, if a selection against UpA dinucleotides existed, the XXU/XXC ratio should be lower when the downstream codon starts with A. Our results indeed confirmed the depletion for CpG, as we observe a significantly higher XXU/XXC ratio upstream a codon starting by a G compared to codon starting with any other base (Figure 3E), avoiding the creation of CpG at the overlap between two codons. Regarding the hypothesis of UpA depletion, the median XXU/XXC ratio upstream a codon starting by A was the lowest of all bases, albeit with a large variance and not different from the values for codons starting by C or U. This result mitigates our previous finding of a slight intra-codon avoidance of the UpA dinucleotide, but is concordant with the overall absence of significant avoidance of UpA within the coding sequence.

### 2.5. Mutational Bias and CpG/UpA Depletion Explain Most of the Variation in Synonymous Codon Usage of Coronaviruses

Having identified GC->AT mutational bias as well as CpG and possibly UpA dinucleotide depletion as variables impacting CUB in *Coronavirinae*, we aimed at further quantifying the differential contributions of each of these factors to the overall CUB variation. We therefore built a linear model quantifying the relative contribution of the different variables (GC3, CpG, UpA, host taxonomy at the level of order, and virus taxonomy at the level of genus) and their corresponding interactions with the covariance of the projections of the first four main dimensions of the PCA for CUB. The analysis of variance demonstrated that variation in GC3, CpG and UpA are, respectively, the best predictors of the first three dimensions of the PCA (Figure 4), respectively explaining 94%, 67% and 56% of the variation in each dimension. By contrast, the integration of host and virus taxonomy stratification levels and their interactions with the compositional variables did not contribute with any further improvement of the model fit to the data. Lastly, the fourth PC dimension was only marginally explained by UpA content and by the *UpA*viral taxonomy* interaction, which explained 19.3% and 18.2% of the PC4 dimension, respectively. Given these results, and considering the power of each PCA axis for explaining variation in CUB, we conclude that variation in GC3 explains around 34%, variation in CpG frequency explains around 14%, and variation in UpA frequency explains around 7% of the total variation in CUB in *Coronavirinae*. On the contrary, the contribution of viral taxonomy and of host taxonomy diversity to explain variation in CUB is negligible.

## 3. Discussion

With the COVID-19 pandemic, the unprecedented wealth of sequence data generated on human SARS-CoV-2 and its close relatives infecting pangolin and bats provides a unique opportunity to investigate the variability and the biological role of synonymous codon usage among *Orthocoronavirinae*. Previous works in the field have been successful at inventorying CUB in SARS-CoV-2 [31,32,58], as well as at reporting the poor match in the observed CUB between SARS-CoV-2 and humans [59]. Together, those studies raised important questions about the nature of the CUB in CoVs, the underlying evolutionary mechanism involved, and their impact on the longer-term evolution of the virus. Here, we explore those questions, by meticulously disentangling the effects that natural selection, dinucleotide avoidance and neutral forces have on the variability in CUB in CoVs.

### 3.1. Mutational Bias and CpG Avoidance Shape Codon Usage Bias in CoVs

We report a strong heterogeneity in CUB among CoVs, mainly driven by GC->AU biased mutations. In analyses restricted to the SARS-CoV-2 genomes, previous studies had identified C->U deamination as the main mutational contributor to spontaneous mutation [50,51,60]. Our work conducted on the largest diversity of CoVs investigated so far is a novel compelling piece of evidence suggesting that mutational bias towards AU is a universal feature among CoVs, regardless of the viral taxonomy and of the host infected. We further report that for all CoVs, C->U substitutions occur at a much higher frequency than G->A substitutions. This asymmetric substitution pattern has been proposed to be driven by a host APOBEC3-like editing process, rather than occurring during virus replication and being associated with mutational biases of the coronavirus RNA-dependent RNA polymerase [51,61]. The *APOBEC3* locus in mammals consists of a series of tandem copies of different *APOBEC3* genes that have undergone a complex evolutionary history of duplications, deletions and fusions, further complicated at the transcriptome level with a large diversity of mRNA [62]. The evolution of the *APOBEC3* locus is tightly linked to viral infections via a different arms race [63,64]. The structure and synteny in the *APOBEC3* locus and the protein repertoire therein encoded are extremely lineage-specific (see for instance [65,66] for Chiroptera or [67,68] for Primates). It is thus important to note that the mutagenic role of host APOBEC3 enzymes acting onto viral genomes and exerting strong evolutionary pressures in a particular virus–host interaction may fuel viral adaptations, which facilitate viral transmission and colonisation of novel host species, as suggested for lentiviruses infecting Primates [69].

In addition to the increased GC->AU substitutions, we remark that CoVs genomes are strongly depleted in the CpG dinucleotide, resulting in the avoidance of synonymous codons and of codon pairs containing this motif. Our results are consistent with previous works demonstrating that the experimental increase in CpG frequencies impaired virus fitness [37,70,71,72]. CpG depletion has been observed across CoVs infecting a whole range of mammals and birds, indicating that the attenuation mechanism might be fundamental to vertebrate eukaryotic antiviral defense, evolutionarily conserved and active over three hundreds of millions of years of evolution [73]. Surprisingly, we did not observe any statistically significant depletion of the UpA dinucleotide in CoV genomes (Figure 3A), while underrepresentation of UpA is a common trend among human RNA virus [57,74]. Analogous to the host immune response against the presence of CpG in the viral genomes, the increased presence of UpA dinucleotides has been formally identified as reducing virus replication [75], potentially through the cleavage of viral RNAs by RNase L [71,76]. Our investigation of the impact of UpA dinucleotide frequency on the synonymous codon frequency displayed mitigated results: avoidance of UpA dinucleotide was observed when the UpA dinucleotide is located at the end of a codon (Figure 3D) but not when the UpA dinucleotide is located at the codon pair boundary (Figure 3E). Hence, we hypothesize that UpA sites in CoVs are at the center of the antagonist effects of mutation bias (promoting the formation of new UpA sites through GC->AU mutational bias) and selection (directly acting against UpA sites). However, testing this mutation-selection trade-off hypothesis goes beyond the power of our bioinformatics analyses and should be properly addressed by actually modifying the UpA content of CoV genomes, quantifying changes in viral fitness and following the evolutionary trajectories of the modified genomes by means of experimental evolution. Altogether, our work fits well with the mounting evidence supporting the biotechnological application of modifications in CpG and UpA dinucleotide frequencies in viral genomes for the production of efficient, safe and evolution-proof vaccines [71].

### 3.2. Lack of Evidence for Translational Selection Acting on CoVs

Mismatches between the mRNA characteristics and the translation machinery can have a strong impact on the quality and quantity of protein production. Selection acting on mutations leading to streamlined translation is known as translational selection. However, studies on the distribution of fitness effect show that the selective advantage associated with beneficial individual synonymous mutations is very low [77,78,79,80,81], implying that translational selection could only be efficient in the case of organisms with large effective population sizes. Indeed, strong phenotypic effects of codon usage on expression levels of single genes have been experimentally reported in some metazoan species with large population sizes (such as drosophila or nematode) [42,82,83]. Similarly, studies in vertebrates have established that translational selection is much weaker in large-sized vertebrates compared to small-sized ones, because the former are slow-growing organisms with a small effective population size [84]. Additionally, the physicochemical organization of vertebrate chromosomes may further hamper translational selection from playing a strong role in vertebrates: chromosomes in many vertebrates are organized in long, consecutive regions enriched in AT or in GC nucleotides, known as isochores [85], so that the factor with the largest individual impact on CUB of vertebrate genes is its physical location along the chromosome [86].

As with cellular genes, viral genes rely completely on the host cell apparatus for translation. It has thus been proposed that selection could act to optimize the match between the virus CUB and tRNA repertoire and abundance in its host, leading to increased efficiency and accuracy of viral protein synthesis [14]. In the case of viruses, the large within-host population sizes that many of them generate during productive infections could allow translational selection to shape CUB. For bacteriophages, CUB has been shown to match codon preferences in the bacterial host [28], which in their turn match the most abundant cognate tRNAs available in the cell [87]. In viruses infecting vertebrates, mounting experimental evidence suggests that modification of viral codon usage leads to sharp changes in viral fitness, the necessary condition for natural selection to act upon viral CUB [88]. However, the results presented here show that host stratification explains only a minor fraction of the global variability of CUB in CoVs, suggesting that for these viruses the variability of synonymous codons is largely unrelated to the type of host infected. Thus, we conclude that translational selection based on a differential tRNA abundance as a function of the host is not the main evolutionary driver of CUB in CoVs. Experimental results on ribosome profiling for *Murine coronavirus* genes have reported that despite the poor match in CUB between this virus and the mouse host, coronavirus mRNAs were translated with similar efficiency as the host mRNAs [89]. Thus, the high frequency of U- and A-ending synonymous codon in CoVs genomes, systematically departing from the CUB of their host [59], would not negatively impact the synthesis of viral proteins.

In metazoans the interplay between the main factors shaping CUB, mutational forces, GC-biased gene conversion and translational selection, have been finely analyzed and interpreted as a function of effective population size [84,90]. Notwithstanding, the same studies identify CUB trends, such as the systematic preference of pyrimidines over purines at the third codon position, for which we still lack an interpretative framework [84]. In our analyses for CoVs, despite the strong explanatory power of variation in nucleotide composition and CpG avoidance to explain variation in CUB, over 40% of the overall variation in CUB still remains to be explained. Given the impact of effective population size on the efficiency of translational selection to shape CUB in metazoans, could translational selection contribute to explain the proportion of the variation in CUB in CoVs that has not been attributed to any variable? Here, we report that the between-host variability of CUB in CoVs accounts for 19.7% of the total variability in CUB, a fraction two times larger than the random expectation. Although statistically significant, this contribution does not necessarily prove that differential adaptation to the translational machinery of the different hosts is shaping CUB in CoVs. Our analyses suggest instead that significant differences in GC3 and UpA occur between CoVs infecting different hosts (Appendix A), so that between-hosts differences in CUB in CoVs could actually also reflect the impact of compositional variation at shaping CUB. A proper answer to the question of the actual role of translational selection in CoVs will require experimental work identifying differences in the natural history of the infection (e.g., target tissue, productivity, clinical presentation) of genetically close viruses upon host switch.

We want to point out a potential limitation for our study, namely that we focused on a nucleotide and CUB analysis for the complete genome, instead of having focused on the individual genes. Furthermore, as detailed in the methods and to prevent biases introduced by annotation errors in the SARS-COV-2 genomes, we limited ourselves to the best-annotated open reading frames, namely ORF1ab, S, E, M and N. These global analyses may have erased any potential signal for differential composition and/or CUB between open reading frames. It is conceivable that different open reading frames in the CoV genomes may present differences in their nucleotide composition and CUB, as is the case in many other viruses [30,91]. This is actually more relevant in the case of CoVs, because of the (+)ssRNA nature of the viral genome. Indeed, transcription in CoVs generates molecules of sub-genomic length, differing in the 5′ starting open reading frame [92], and among all proteins potentially encoded by a CoV genome or sub-genome, only the one corresponding to the open reading frame located in the 5′ end is actually translated [93,94]. Thus, in CoVs, the central mechanism for gene expression regulation is transcriptional regulation, which may evidently facilitate, for instance, the selection for different coding trends between open reading frames located in the 5′ and in the 3′ ends of the viral genome. This question can be properly addressed by differentially analyzing the translation efficiency of each viral open reading frame during the course of the individual cell infection, relating them to the putative CUB differences between the proteins encoded in the genome 5′ and involved in the replication-transcription complex on the one hand, and the structural proteins encoded in the genome 3′ on the other hand.

### 3.3. Composition of CoVs Genomes Tends to Reach Their Mutational Equilibria

In order to investigate mutational bias in CoVs and to disentangle it from the effect of natural selection, we analyzed mutations occurring during recent evolution in 16 viral metapopulations at different taxonomic host and viral levels. With this approach, we aimed at comparing mutational biases, while minimizing the impact of natural selection. When considering highly accurate folded site-frequency spectrum at the population level, we observe that endemic hCoVs are closer to an equilibrium in the AU—GC allele distribution than zoonotic, recently acquired hCoVs (Figure 2A,B). This suggests that endemic hCoVs may have undergone a compositional drift towards a novel equilibrium under the novel mutational pressures in the human upon host switch. Indeed, at the short time scale that our analyses can explore for the SARS-CoV-2 epidemics in humans, we verify a small albeit significant trend towards GC reduction (Figure 2C). Similar trends have also been reported when analyzing different datasets [95,96]. Future work will benefit from accessing a sufficient number of SARS-CoV-2 non-human strains collected from different hosts (bats and pangolins in first instance, given the current knowledge) to assess the population-level characteristics of the direction and intensity of mutational bias in endemic host species. Additionally, endemic hCoVs exhibit a large variation in observed synonymous GC3-content, ranging from 18.7% to 30.6%, which suggests that the strength of the mutational bias causing C->U changes is not similar among all hCoVs. We hypothesize that these differences in mutational signature could be related to the differences in the host mutator APOBEC3 repertoires, which vary between hosts, but also between cell types within an organism, so that changes in the host tropism will have an impact on the actual mutational intensity and direction that the viral genomes experience. It is interesting to note that when performing a similar analysis for non-human CoVs, a number of viruses display a significant departure from neutrality in SFS skewness, mostly flagrant for avian gamma-coronavirus *Infectious bronchitis virus* as well as for the SARS-CoV-2 sequences from American mink *Neovison vison* isolates, but also evident in MERS sequences from camel isolates, *Camel alphacoronavirus* and *Porcine coronavirus HKU15*. Should our interpretation of the SFS skweness for hCoVs be correct, we could speculate that these viral lineages might have undergone a recent shift in mutational bias, compatible with a host switch or tropism shift event.

### 3.4. Conclusion: CUB Is a Poor Proxy to Predict Zoonotic Infection in CoVs

SARS-CoV-2 is the third *Coronavirinae* zoonotic spillover of the 21-century, and the associated COVID-19 poses an unprecedented burden on human public health. Understanding the drivers and facilitators of interspecies viral transmission in CoVs is a key public health and fundamental research priority. Nucleotide composition and CUB are distinctive characteristics of a virus, shaped by the global action of different pressures, including host adaptation [29,97,98]. In the case of SARS-CoV-2, the viral CUB is closer to that of humans than other hCoVs [58]. Although one could interpret that this “closer-to-human” CUB may have facilitated the establishment in humans following the zoonotic transmission event, our overall results show that the particular match between CoVs and host CUB alone plays a minor role in the chances for a zoonotic spillover to establish in humans. Instead, we have shown here that the main drivers of CUB (low GC3 content, strong CpG avoidance and slight UpA avoidance) are common to all CoVs, independently of viral taxonomy and of the hosts they infect. We interpret thus that the role of CUB at modulating the chances of a CoV zoonotic spillover to thrive in humans is negligible. Notwithstanding, the available data suggest that upon colonization of humans, endemic hCoVs have experienced a compositional drift towards a novel compositional equilibrium in the new hosts, and that this could also be the case for SARS-CoV-2 if the current pandemic evolves towards an endemic circulation in humans.

## 4. Material and Methods

### 4.1. Data Collection and Processing

To access the diversity of coronavirus species we downloaded sequences classified within the *Coronaviridae* families from the virus–host database (https://www.genome.jp/virushostdb/, accessed on 19 March 2021). Because the 212 viral entries present in the database are composed of a mixture of sequence classified at different taxonomy levels, i.e., species (HKU1, HKU4, HKU5, …) or strains (HKU4-1, HKU4-2, HKU4-3, …), we used CD-HIT (option–c 0.95–n 5) to cluster together sequences sharing over 95% of identify. A total of 82 distinct groups of complete genomes were identified, from which we manually choose one single representative member, preferentially selecting the NCBI manually curated genomic sequence.

For the intra-species diversity, we downloaded strain sequences from the Virus Pathogen Resource database (VIRP). Unusually long or short sequences (>130% or <70% of the median length of the reference sequence of a species) were filtered out. In addition, the complete genomic sequences of SARS-CoV-2 isolates were obtained from GISAID (available at https://www.gisaid.org/epiflu-applications/nexthcov-19-app/), accessed on 6 May 2020 to collect the sequences for our metapopulation analysis. SARS-CoV2 sequences bellow 29,000 bp were automatically discarded. Detailed accession ID for both datasets are provided in Appendix A.

### 4.2. Nucleotide Composition Analysis

From the NCBI Genbank files of the previously selected 82 complete genomes, we used biopython to concatenate the coding sequences of each CoV genome, and then compute the total synonymous codon frequencies for each genome. Only ORFs from the orf1ab, spike glycoprotein (S), envelope small membrane protein (E), membrane protein (M), and nucleoprotein (N) were considered for our downstream analysis, as those ORFs were presumably the best annotated ones across CoVs. We want to raise a word of caution, as we have analyzed the complete coding genomes of the different viral taxa without individual focus on the different genome regions that may be differentially transcribed and replicated and present local peculiarities in nucleotide composition and/or synonymous codon usage. Thus, our results refer to overall trends for the complete genomes and do not exclude the possibility of the presence of local compositional anomalies. A matrix of 82 (CoVs) × 59 (synonymous codons) was created, which served as input for either our PCA analysis (FactoMineR) and correspondence analysis (run with R package ade4). In parallel, we calculated the dinucleotide observed/expected abundances in coding sequences by calculating the ratio rXY = fXY/fXfY, where fXY denotes the observed frequency of the dinucleotide XY and fXfY the product of the individual frequencies of the nucleotides X and Y in a sequence. We used as significant lower and upper boundaries the thresholds of 0.78 and 1.25, corresponding to *p* < 0.001 for sufficiently long (>20 kb) random sequences [99].

### 4.3. SNP Calling

To identify intra-species polymorphisms, we used mummer [100] to perform a genome-to-genome alignment of our 16 metapopulations against its corresponding reference sequence, to then call SNPs using the program show-snps from the mummer tools suite. Multi-allelic sites and structural variants were filtered out to only consider bi-allelic variants. In order to differentiate synonymous to non-synonymous mutations, using biopython we annotated each variant based on the CDS annotation provided within the corresponding Genbank reference sequence.

### 4.4. Assessment Mutation Profiles

For the SARS-CoV2 genome, accessing the mutation profiles at synonymous positions is trivial since one SARS-CoV2 sequence sampled from the early pandemic has been sequenced. Hence, to estimate the whole-genome nucleotide flux, we need to count the frequency of each type of mutation with respect to this quasi-ancestral genome. However, for the other CoVs genomes, such a sequence has not been characterized yet, and we need to develop a strategy to determine which allele is ancestral and which is derived. Here, we used a conservative approach, considering variants with a minor allele frequency lower than 10%, with the ancestral state was considered for the major allele while the minor allele was considered as the derivate state.

### 4.5. Site Frequency Spectrum Assessment Mutational Equilibrium

Prior to the inference the site frequency spectrum (SFS), the population structure of each metapopulation was characterized. Using biopython, we first transformed our multi-sample VCF file into a multi-alignment (MSA) file, which served as input file for the identification of population sub-structure and recombinant sequences using fastGear [101]. FastGear presents the advantage of identifying population genetic structure from an alignment, as well as detecting recombination between the inferred lineages. Complete genomes presenting long recombining segments (>200 bp) were removed from each dataset, while short recombination segments were masked for the downstream analysis.

From the VCF file of each metapopulation, the allele frequency of each variant was computed using vcftools (--freq). In order to work in a framework where population structure of our metapopulation does not interfere with the shape of the derived allele frequency counts, we focused for each viral entry on the single lineage per metapopulation, having the highest number of complete genomes (as previously identified by fastGear). For each of the 18 viral metapopulations we calculated the derived allele frequency spectrum by comparing each extant sequence with the ancestral one. Previous studies have shown that the analysis of the derived allele frequency spectrum are sufficiently robust to demographic and/or sampling effects [102,103]. We constructed then the folded SFS by polarizing the derived allele frequency as AU-enriching or GC-enriching with respect to the ancestral sequence, as in [56] using the R script kindly provided by Dr. S. Glémin. The folded SFS is symmetrical if and only if GC content is at mutation balance equilibrium, and this result is robust for populations outside demographic equilibrium [56]. Indeed, analyses of the folded SFS can be used to infer demographic histories, reflecting bottlenecks and population expansions (see for instance [104]).

## Figures and Tables

**Figure 1 viruses-13-01800-f001:**
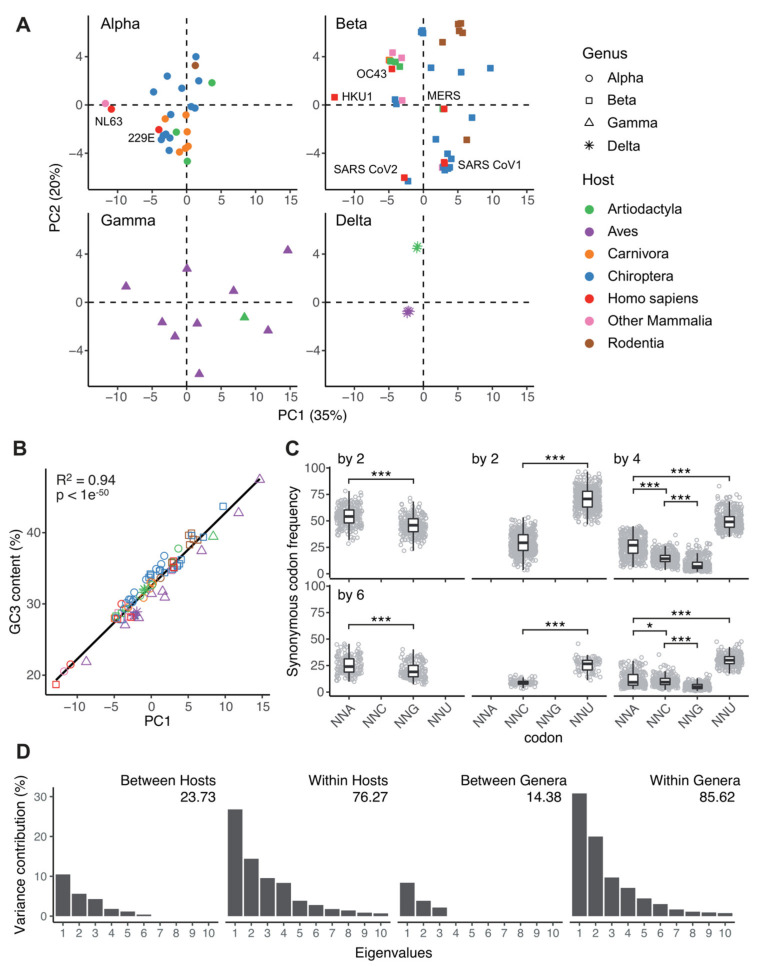
Variation in synonymous codon usage in *Coronavirinae*. (**A**) Principal component analysis of synonymous codon usage among CoVs. Each dot corresponds to an individual CoV, for which the frequencies of the 59 synonymous codons have been calculated. CoVs are stratified after the corresponding genera (Alpha-Delta), and host (color code in the inset). The percentage of the variance explained by the first and second axes is given in parenthesis. (**B**) Correlation between the projection on the first of principal components shown in (**A**) and the average GC-content at third codon position (GC3) for the corresponding viral genome. Symbols and color code are the same as in (**A**). The results for a Pearson’s correlation test are shown in the inset. (**C**) Synonymous codon usage frequency stratified by amino acids of multiplicity 2, 4, or 6 synonymous codons. Differences in median codon usage frequency values were assessed by a paired Wilcoxon signed rank test (code for *p*-value: * <0.05; ** <0.01; *** <0.001). (**D**) Results of an internal correspondence analysis for the contribution of the levels “viral taxonomy” (four levels, corresponding to viral genera) and “host taxonomy” (fourteen levels, corresponding to host orders) to explain variability among the eigenvalues of viral genomes on the first 10 principal components in the PCA shown in (**A**). The values for expected explanatory power for a homogeneous distribution were and 3.72% (*p*-value < 9 × 10^−4^) for viral taxonomy and 7.59% (*p*-value < 9 × 10^−4^) for host taxonomy.

**Figure 2 viruses-13-01800-f002:**
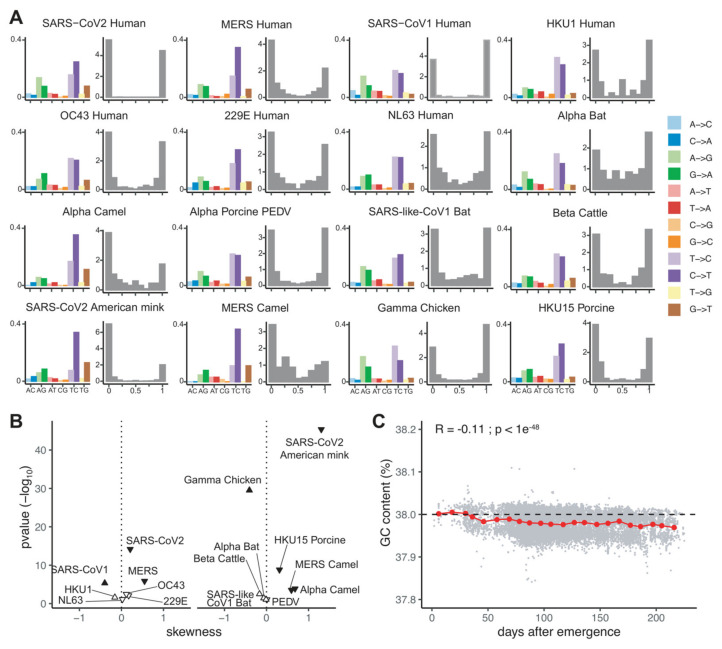
Mutational bias and equilibrium among endemic and recently zoonotic coronaviruses. (**A**) (left side of each panel) Relative substitution rates for all twelve transition and transversion events, calculated from the total number of SNPs called on the synonymous compartment of a metapopulation composed by all available sequences of the corresponding virus, curated, filtered and aligned. In the x axis, for each of the nucleotide pairs indicated, the frequency of the two substitutions are indicated, e.g.,: A->G in light green and G->A in dark green for the AG pair, or T->C in light purple and C->T in dark purple for the TC pair. (right side of each panel) Folded Site Frequency Spectrum (SFS) of G/C-to-A/T substitutions. For each viral metapopulation, the frequencies of SNPs involving a change in GC content called are plotted, the *x*-axis progressing from GC-enriching to AT-enriching substitutions. (**B**) Volcano plot of the folded SFS skewness values presented in (**A**), plotted against the probability that such a value could have been obtained if the corresponding metapopulation were at its mutational equilibrium. The left panel displays CoVs infecting humans while the right panel displays CoVs infecting other mammalian hosts. Downward and upward triangles, respectively, refer to SFS-negative and SFS-positive skewness values. Filled triangles indicate metapopulations with a skewness significantly different from the expected one at mutational equilibrium. (**C**) Temporal evolution of the SARS-CoV-2 GC-content in the coding genome during the spread of the pandemic. The GC-content of each individual SARS-CoV-2 genome is represented by a gray dot. Red dots represent the average GC-content by time window of 10 days. The results of a Pearson correlation test of GC content over time for the complete data set (n = 81,963) are displayed in the inset.

**Figure 3 viruses-13-01800-f003:**
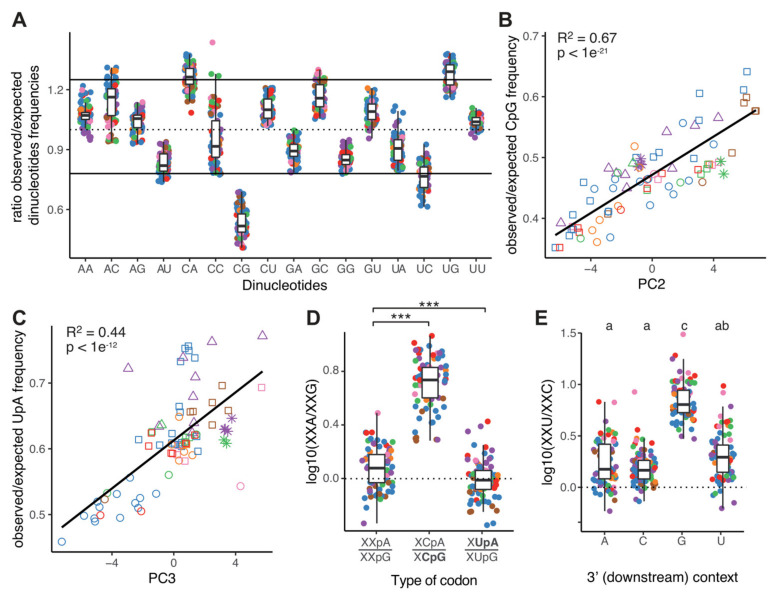
Dinucleotide Bias in *Coronavirinae.* (**A**) Box (median, 1st and 3rd quartiles) and whiskers (95% CI) plot for the ratio of the observed over the expected frequency (dinucleotide relative abundance) for all 16 dinucleotides in the coding sequences of 82 CoVs. Continuous lines indicate the thresholds for considering over/under representation of the corresponding dinucleotide over the expected values given the frequencies of the individual nucleotides. The threshold values at 0.78 and 1.25 have been phenomenologically determined following [57]. Colors correspond to the different host taxonomy orders. (**B**) Linear regression between the projections of each individual viral genome on the second dimension of the synonymous codon usage Principal Component Analysis (Figure 1A) and the observed/expected frequency of the CpG dinucleotide. Shapes correspond to viral taxonomy genera and colors correspond to host taxonomy orders. (**C**) Linear regression between the projections of each individual viral genome on the third dimension of the synonymous codon usage Principal Component Analysis and the observed/expected frequency of the UpA dinucleotide. Shapes correspond to viral taxonomy genera and colors correspond to host taxonomy orders. (**D**) Box (median, 1st and 3rd quartiles) and whiskers (95% CI) plot for the ratio in synonymous codon frequency of A- over G-ending codons, calculated for codon families with multiplicity two lacking CpG and UpA dinucleotides (Gln, Lys and Glu, indicated as “XXpA/XXpG”, constituting the reference set for comparison), amino acids encoded by CpG-ending codons (Ser4, Pro, Thr, Ala, indicated as “XCpA/XCpG”), and amino acids encoded by UpA-ending codons (Leu2, Leu4, Val, indicated as “XUpA/XUpG”). Colors correspond to the different host taxonomy orders. Differences within amino categories were assessed by a paired Wilcoxon signed rank test (Bonferroni correction, code for *p*-value: *** <0.001). (**E**) Box (median, 1st and 3rd quartiles) and whiskers (95% CI) plot for the ratio in synonymous codon frequency in U- over C-ending codons, calculated for codon families with multiplicity four, and stratified by the nature of the 3′ downstream nucleotide. Differences depending on 3′ base context were assessed by a paired Wilcoxon signed rank test (Bonferroni correction) and summarized among sets of groups statistically different one from another. Colors correspond to the different host taxonomy orders.

**Figure 4 viruses-13-01800-f004:**
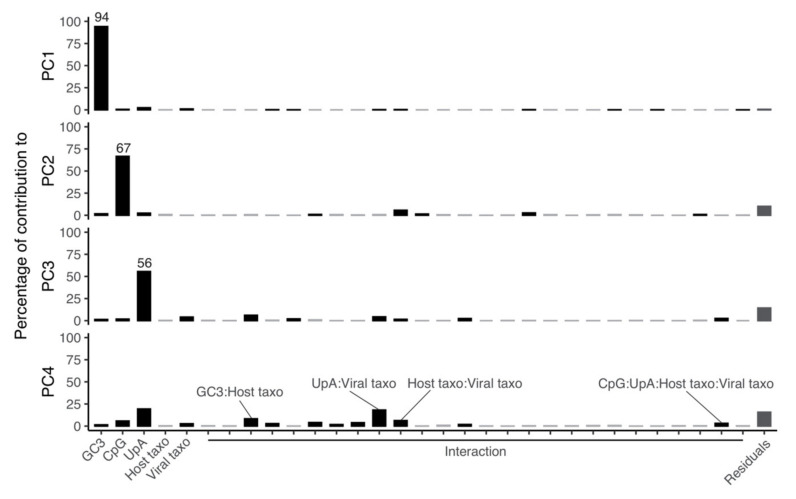
Linear model quantifying the relative contribution of the different variables to the projections of the first four main dimensions of the synonymous codon usage PCA (Figure 1A). Variables included in the linear model are: GC3-content, observed/expected frequency of the CpG dinucleotide, observed/expected frequency of the UpA dinucleotide, host taxonomy at the level of family, viral taxonomy at the level of genus, and all their pairwise interactions. Statistically significant contributions are shown as black bars, stating the percentage of the total explanatory contribution to the correspondent PCA dimension.

## Data Availability

We acknowledge the three different sources of genomes that we employed: the virus–host database (https://www.genome.jp/virushostdb/, version of 19 March 2020), the Virus Pathogen Resource database (VIRP, https://www.viprbrc.org/brc/home.spg?decorator=vipr, version of 26 April 2020), and the GISAID database (https://www.gisaid.org/, version of 19 August 2020).

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
