# Peer review of "Variability in Codon Usage in Coronaviruses Is Mainly Driven by Mutational Bias and Selective Constraints on CpG Dinucleotide"

_viruses, 2021, doi:10.3390/v13091800_

Round 1

Reviewer 1 Report

General

This article reports on genetic variation of several animal coronaviruses including human SARS-CoV-2 causing the Covid19 disease. The authors analysed, in detail, mutation trends in 16 groups of betacoronaviruses totally encompassing 82 genomes. They found that the mutation patterns are similar across the groups indicating similar evolutionary trajectories in different viruses and their hosts. They further revealed that CpG depletion apparently caused by mutation biases is a driver of amino acid codon preferences in coronavirus genomes.  The methods used are adequate, The results are adequately interpreted. There are no technical and interpretation flaws. This is a strong study. I have only minor comments as below.

Major issues

1. The codon is composed of three nucleotides, a dinucleotide (e.g. CG) has three possible locations. Herewith, they can be designated as (CG)12, (CG)23 and (CG)31 respectively. While the position (CG)23 has been analysed in detail the other two apparently not. The (CG)12 and (CG)31 occur in both synonymous and non synonymous sites while the (CG)23 mostly in synonymous sites. I believe that comparisons of CpG depletion between all three locations would be informative since it may to what extent mutation biases/asymmetries contribute to amino acid changes. While I do not insist on the experiment since it may require extensive computation it should be at least discussed.

2. Figure 2 – X axis. In legend the meaning of individual columns should be explained, e.g. A>C, C>A frequencies are in light and dark blue, respectively. Maybe that description of the columns should already appear in the first row.

3. Line 579 and methods. Concatenated sequences containing five different subregions (orf1ab, spike glycoprotein (S), envelope small membrane protein (E), membrane protein 581 (M), and nucleoprotein (N)) of CoV genomes were used for the CUB analysis. The danger of such an approach is that individual subregions may have different nucleotide composition and subjected to different selection pressures. For example, the G+C content of the SARS-CoV-2 N protein is 47% while the rest of the genome  is about 37% only. The unequal nucleotide distribution is apparently reflected in variability in the dinucleotide content. The N protein has more CpGs (0.031 frequency) than the rest of the genome (about 0.015). There is even a small CpG island present towards the 5’ end showing no CpG depletion. For such reasons, I feel that generalisation of statements should be weaken.   

4. I miss a broader discussion about the cause of CpG depletion in coronavirus genome. For example, why is CG but not GC, GG or CC taken as the target for mutation? Few hints could be considered:

-Line 404. The sentence “…, we remark that CoVs genomes are strongly depleted in the CpG dinucleotide, resulting in the avoidance of synonymous codons and of codon pairs containing this motif.“ This could also be put other way round. The CpGs particularly those at the 2,3 positions of codons are mutated because of their highest occurrence in synonymous codons out of all dinucleotides. These are TCG, CCG, ACG and GCG which code for serine, proline, threonine and alanine, Mutation of G at codon position 3 into T, C or A in all of them does not change the amino acid they encode.In other words CGs at 2,3 mutate because of low chance of inducing a deleterious mutation compared to other dinucleotides.

-Mechanistically, loss of CpG can be explained by a dual contribution of mutation biases. In FIgure 2, the most frequent substitutions are C>U and U>G at least in SARS-CoV-2. The G>A substitutions are the third.  First, overall C deamination trend leading to CG>UG. Second, G>U transversion would lead to CU dinucleotides. Perhaps, G>U transversions which seem to variable across the genomes (Figure 2) may, in part, explain differences in the CpG content between viruses (Figure 3).

5. Line 594. It is unclear what the reference sequences were. Perhaps, a representative genome for each of the 16 population? Please, explain.

6.Data from the first year of the Covid-19 pandemic revealed that emerging mutations in the SARS-CoV-2 genome are non-random and highly skewed to C>U substitutions. Some of the studies were using data from the early stage of the pandemics while the others from more advanced stages. Some studies demonstrated that the C>U mutations are not limited SARS-CoV-2, but also in related animal viruses. Finally, there seems to be no difference in mutation profiles between viruses isolated throughout the pandemics suggesting that mutation biases are pretty much stable over time. Because the C-deamination mediated mutations seem to be responsible for mutations biases in codons as demonstrated by this study I feel that the following references should be considered for citation:

Roy, C.; Mandal, S.M.; Mondal, S.K.; Mukherjee, S.; Mapder, T.; Ghosh,W.; Chakraborty, R. Trends of mutation accumulation across global SARS-CoV-2 genomes: Implications for the evolution of the novel coronavirus. Genomics 2020, 112, 5331–5342.

van Dorp, L.; Acman, M.; Richard, D.; Shaw, L.P.; Ford, C.E.; Ormond, L.; Owen, C.J.; Pang, J.; Tan, C.C.S.; Boshier, F.A.T.; et al. Emergence of genomic diversity and recurrent mutations in SARS-CoV-2. Infect. Genet. Evol. 2020, 83, 104351.

Matyášek, R.; Kovarík, A. Mutation Patterns of Human SARS-CoV-2 and Bat RaTG13 Coronavirus Genomes Are Strongly Biased Towards C>U Transitions, Indicating Rapid Evolution in Their Hosts. Genes 2020, 11, 761.

Vankadari, N. Overwhelming mutations or SNPs of SARS-CoV-2: A point of caution. Gene 2020, 752, 144792.

Minor issues

Line 592 „intra-specie“ should read „intra-species“

Line 701 „calucated“ should read „calculated“

I understand that T is used here instead of U for simplicity while it should be explained at its first appearance stating that the coronavirus genomes contains U and not T nucleotides.

Author Response

Here is the answer to R1.

Reviewer 2 Report

Great paper and a very enjoyable read. 

Some small comments for consideration.

Materials and methods could benefit from some description of “folded site frequency spectrum (SFS)” although described and referred to in the introduction.

Line 55: Ikemura 1985a has a «a» in the references but there is no «b». 

Line 193: the number of genomes vary greatly between different metapopulation (from 9 to 9588). It could influence the number of SNPs discovered. The authors indicate that the number of SNP is highly comparable. Does that mean that there is no significant relationship between the number of genome included and the number of SNPs? Did they do a statistical test to check that?

Line 421-422: The figure 4D and 4E don’t exists, I think the authors means 3D and 3E?

Line 516: It states that the COVs were found at their GC equilibrium and site figure 2C, however it is said previously (L. 269-271), citing the same figure, that the GC content is significantly decreasing.

Line 523: Are the authors citing the right figure?

Line 541-542: “Jenkins and Holmes 2003” and “Cristina et al. 2015” are not cited in the reference. Check references.

Author Response

Here is the answer to R2.

Reviewer 3 Report

This is a very interesting work on the theme of the interaction (if any) between the codon bias of an infecting virus and that of the infected host.
The paper is of very good quality, well written and of a clear and terse line of the argument. 
The study aims at a synchronic characterisation of the CUB of several CoVs. The main, interesting and somehow unespected, result is that CUB of the viruses seems to be largely independent from that of the hosts and is poorly affected by natural selection. In other words the codon bias of the virus and that of the host are evolutionarily uncoupled and the codon bias of coronaviruses appears to be under a general mutational bias. The methods the data analysis and the discussion are sound and may remarkably contribute to this line of research. Let me just suggest one remark that could explain the somehow negative results here shown. The authors consider each genome per se, and the main observable they elaborate on are the overall nucleotide frequencies in each viral genome. The interesting point is that in most viral genome the CUB seem to correspond to a steady quasi-neutral state, whereas in the SARS species, actually interacting with humans, is quite plausible that CUB is in an evolving adaptive state. So, it would be nice to look at some CUB metric dynamically, using the data of the GISAID databank. Moreover, I have the impression that averaging over the different viral genes could explain why the translation-selection hypothesis is blurred. I would be very interested in looking at the different trajectories of the CUB in the different viral proteins. In other words I would suggest to the authors make some comment and observation about differences in nucleotide frequencies between the genes of different viral coronavirus proteins. 

Minor point, the non parametric test used. e.g. in supplementary figure 8 is misspelled: Krustal-Wallis for Kruskal-Wallis.

Author Response

Here is the answer to R3.
